# Rugby game performances and weekly workload: Using of data mining process to enter in the complexity

Romain Dubois[1,2]*, Noëlle Bru[3], Thierry Paillard[1], Anne Le Cunuder[4], Mark Lyons[5], Olivier Maurelli[6], Kilian Philippe[1,4], Jacques Prioux[4]

1 Laboratoire Mouvement, Equilibre, Performance, Santé (MEPS, EA-4445), Université de pau et des pays de l'Adour, Tarbes, France, 2 SASP Club Athlétique Brive Corrèze (CABC), Brive-La-Gaillarde, France, 3 CNRS / University of Pau & Pays de l'Adour/ E2S UPPA, Laboratory of Mathematics and its Applications of Pau – MIRA, UMR5142, Allée du Parc Montaury, Anglet, France, 4 Movement, Sport and Health Laboratory (EA-1274), Faculty of Sport Science, Rennes, France, 5 Biomechanics Research Unit, Department of Physical Education and Sports Sciences, University of Limerick, Limerick, Ireland, 6 Adaptations Physiologiques à l'Exercice et Réadaptation à l'Effort (EA-3300), Université de Picardie, Amiens, France

* romain.dubois55@orange.fr

**Data Availability Statement:** Thank you for the players who participated to this study. The workload parameters were the propriety of the players and for this study, they were accepted that

## Abstract

This study aimed to i) identify key performance indicators of professional rugby matches, ii) define synthetic indicators of performance and iii) analyze how weekly workload (2WL) influences match performance throughout an entire season at different time-points (considering WL of up to 8 weeks prior to competition). This study uses abundant sports data and data mining techniques to assess player performance and to determine the influence of 2WL on performance. WL, locomotor activity and rugby specific actions were collected on 14 professional players (26.9 ± 1.9 years) during training and official matches. In order to highlight key performance indicators, a mixed-linear model was used to compare the players' activity relatively to competition results. This analysis showed that defensive skills represent a fundamental factor of team performance. Furthermore, a principal component analysis demonstrated that 88% of locomotor activity could be highlighted by 2 dimensions including total distance, high-speed/metabolic efforts and the number of sprints and accelerations. The final purpose of this study was to analyze the influence that WL has on match performance. To verify this, 2 different statistical models were used. A threshold-based model, from data mining processes, identified the positive influence (p<0.05) that chronic body impacts has on the ability to win offensive 1 on 1 duels during competition. This study highlights practical implications necessary for developing a better understanding of rugby match performance through the use of data mining processes.

## Introduction

Rugby union (RU) became a professional sport in 1995 and has since come across multiple ethical and financial issues. The incessant increase in game intensity and competitive demands

these workload parameters were used to better understand the links between the workload and the performance. These data can be made public from June 2021. Nevertheless, anticipated data access from qualifying researchers with an approved protocol will be possible with the agreement of the club (SASP Club Athletique Brive Correze) by contacting the corresponding author (romain. dubois55@orange.fr), the club (contact@brive-rugby.com) or the head S&C coach during the study period (s_pollyfr@yahoo.fr).

**Funding:** No funding was received for this study and there was no conflict of interest for this study. The SASP Club Athletique Brive Correze provided us access to the GPS and activity data, but it had no role in study design, data collection and analysis, decision to publish, or preparation of the manuscript.

**Competing interests:** We have the following interests: Romain Dubois was employed by SASP Club Athletique Brive Correze. There are no patents, products in development or marketed products to declare. This does not alter our adherence to all the PLOS ONE policies on sharing data and materials, as detailed online in the guide for authors.

(i.e. promotion-relegation championships), among other factors, has greatly contributed to enhanced risks of injury and non-functional adaptations [1,2,3]. Optimizing physical preparedness has, therefore, become the main concern for team staffs. Workload (WL) monitoring, its' management and developing optimal adaptational capabilities are important parameters to consider in elite team-sport environments [4,5]. Indeed, many studies demonstrate the influence of weekly workloads (2WL) on acute and chronic physical performance, physiological adaptations and injury risks in elite rugby players [6,7].

In spite of the correlations between WL, performance, and more particularly physical performance, which are commonly accepted in team sports, very few studies have successfully established these relationships in a competitive context [8,9]. One of the main reasons certainly resides in the difficulty to identify and evaluate the key performance indicators (individual and collective) in team-sports. Nevertheless, for some time now, various studies succeed to reveal some tactical, technical and physical key performance indicators during RU games at different age categories and level of play [10,11,12]. Furthermore, elements of research outline some individual technical skills as being directly correlated to playing performance. Ortega et al. [13] and Den Hollander et al. [14] demonstrated that the percentage of successful tackles, the amount of defensive line breaks and the number of offensive duels won (tackle breaks) positively influenced individual and team performance during RU matches.

Other reason for the complexity of studying WL and its' effect on game performance is the elaboration of a valid and reliable longitudinal monitoring protocol (training and competition). Indeed, according to Fernandez et al. [15], "Physical performance has not yet taken much attention from the research community, due to the difficulty of accessing this information with the same devices during training and competition". For a few years now, microtechnology (GPS and inertial sensors) used in rugby has monitored activity during training and matches with acceptable accuracy [16]. Novel technology has provided the possibility of collecting sport specific data; individual (internal and external parameters) and team (match analysis) statistics. These tracking means provide staff with a large amount of data to analyze. Appropriate modelling of training WL and performance (in a competitive context) is necessary to give a practical meaning to this data [17].

The main objective of this study is to demonstrate how WL influences game performance (individual and collective) in short and moderate terms during a professional RU season. However, as mentioned above, studying the relationships between WL and match performance requires preliminary steps. Hence, the intermediate objectives will be i) to identify key performance indicators during professional RU matches, ii) to elaborate synthetic indicators of performance as to facilitate data analysis, iii) and finally to analyze the influence of 2WL on changes of match performance during an entire season.

## Material and methods

### Participants

Fourteen professional RU players (6 forwards and 8 backs) (age: 26.9 ± 1.9 years; height: 185 ± 7.9 cm and weight: 97.6 ± 13.2 kg) volunteered to participate in this study. All players had been playing professionally for several years (experience: 137.1 ± 73.4 professional matches) and were active members of the same team (CA Brive Correze which took part at the 1$^{st}$ professional division of French championship—Top 14). All subjects gave informed consent to participate in the experiment in accordance with the Declaration of Helsinki. The study protocol was conducted with the support of medical and technical staffs of the professional team. Finally, the study respected the ethical guidelines of the Rennes university and research laboratory associated at this study.

## Procedure

WL and match activity of 14 players were monitored throughout a professional RU season. WL parameters were obtained from different methods (S-RPE, heart-rate (HR) based methods, and GPS tracking). WL parameters were analyzed with different weekly rolling averages (up to 8 previous weeks). Rugby match activity was assessed by GPS tracking (locomotor activity) and completed with video analysis to identify sport-specific activity (tackle count, duels won, . . . See Table 1). Team performance (victory vs defeat and positive vs negative) was analyzed to highlight the key performance indicators during elite RU matches. Data mining and data mining processes were used once data collection was completed. This strategy was elaborated to identify key performance indicators and to underline the influence of WL parameters (acute and chronic) on RU performance.

## Raw data collection

The general organization and WL distribution during this season was presented in a previous study [6]. The season lasted 48 weeks including 8 pre-season microcycles. The competitive phase (40 weeks) contained 32 official matches. To reach the objectives of this present study, internal and external WL were quantified during training and matches. During matches, performance and physical activity were assessed by a microtechnological system (SPI-HPU, 5 Hz, GPSport, Australia), and though video analysis. Video analysis was used to record rugby-specific activity: attempted tackles, successful tackles, defensive line breaks, ruck participation, etc (Table 1). Team performance was identified from match results (victory vs defeat) to which

**Table 1. Listing of specific actions recorded and qualified by video analysis.**

| Actions | Abbr | Units abbr | Description |
|---|---|---|---|
| Tackle attempted | T | n | Number of tackles attempted during the match. |
| Successful tackle | ST | n | Number of tackles completed, when the players block the opposite player who carried the ball. |
| % of tackle successes | % ST | % | The percentage of tackle completed. |
| Offensive tackle | OT | n | Number of tackles where the player, in defensive context, pushes back the ball carrier. |
| Ruck participation | Ruck | n | Number of times where the player arrived in ruck to allow the attacking team to conserve the ball. Only the 3 first ruck participants were count. |
| % of ruck participation | % Ruck | % | Ruck participation relativized to the number of rucks performed by all the team. |
| Number of balls played | BP | n | Number of balls played by the player. |
| Meters won | M won | m | Number of meters covered by the ball carrier in direction to the try line (only when the player gains grounds). |
| Offensive dual won | ODW | n | Number of times where the players beat the defender in breaking the tackle. |
| Line break | LB | n | Number of times where the ball carrier breaks the defensive line. |
| Penalty | Pen | n | Number of penalties conceded by the player, signaled by the referee of the match. |
| Activity score | Act Sc | n.min$^{-1}$ | Number of specific actions mentioned above relativized to the ball-in-play time. |

n: number; min: minutes; %: percentage, m: meters, n.min$^{-1}$: number of actions by minute.

**Table 2. Parameters used to quantify the workload during training phases.**

| Parameters | Abbr | Units | Type | Description |
|---|---|---|---|---|
| Sessions RPE | S-RPE | AU | Internal | WL quantification method obtained in multiplying the intensity of training (CR-10 scale) by the volume of training [19]. |
| Volume | Vol | h | External | Number of practice hours (trainings and games) during the week. |
| Total distance | TD | m | External | Assessed from GPS technology, corresponds to the total distance covered by the players during the training and/or the matches. |
| High-speed running | HSR | m | External | Sum of the distance covered above 14.4 km.h$^{-1}$ [20]. |
| High-metabolic power distance | HMPD | m | External | Sum of the distance covered above 20 W.kg$^{-1}$ [20]. |
| Sprint distance | Sp Dist | m | External | Sum of the distance covered above 25 km.h$^{-1}$. |
| Sprint Number | Sp N | n | External | Number of times the player has run more than 25 km.h$^{-1}$. |
| Accelerations | Acc | n | External | Number of accelerations performed above 2.5 m.s$^{-2}$. |
| Sprint and accelerations | Sp+Acc | n | External | Number of sprints (>25 km.h$^{-1}$) and high-accelerations (>4 m.s$^{-2}$). |
| Severe and high impacts | HI | n | External | Number of impacts measured by inertial captors, with an intensity greater than 8G. |
| New Body Load | NBL | AU | External | Manufacturer indicator calculated from accelerometer data aiming to reflect both the volume and intensity of these accelerations in three planes (X,Y,Z). |
| Training impulses | TRIMPS | AU | Internal | HR-based method to evaluate WL during training [20]. |
| Low heart rate effort | LHRE | min | Internal | Time spent under 85% of HR$_{max}$. |
| High heart rate effort | HHRE | min | Internal | Time spent above 85% of HR$_{max}$. |

AU: Arbitrary units; h: Hour; m: meters; n: number; min: minutes; WL: workload; HR: heart rate; HR$_{max}$: maximal heart rate

was added another type of classification (cf. "Britannic Ranking", see below) which considers the influence of match location on results [18].

Workload quantification: Throughout the season, WL was quantified during each training session using different monitoring methods: session-RPE (S-RPE = RPE (CR-10 Scale) x session duration (expressed in min)) [19], HR-based methods (i.e, TRIMPS; Polar T34, Polar Electro, Finland). External WL was assessed with the use of electronic performance and tracking systems which included GPS and microsensor technology (accelerometers, gyroscopes and magnetometers). 2WL was defined as the sum of WL of each session included in the microcyle (in the present case, all matches were held on Saturdays and a 1-week microcycle corresponds to a Monday-Sunday working week) [6]. The different parameters used to analyze WL during training are specified in Table 2.

Locomotor activity and performance during matches: During the 32 official matches of the season, locomotor activity of players was tracked by microtechnology using the same parameters than those used during training. HR recordings were nevertheless different between matches and training (Table 2). Additional rugby specific actions were recorded by video analysis. This permitted quantification and qualification of rugby specific actions. A qualified video analyst was responsible for collecting data for each rugby match. The specific actions analyzed during the matches are presented in Table 1. In order to accurately normalize data, GPS data and sport specific actions were expressed relatively to playing time. Data corresponding to less than 10 min of playtime was not used for this study. In the aim to focus on individual variations (and to remove inter-individual differences from the performance potential), a Z-score specific to each player was calculated for all performance and locomotor activity parameters. This Z-score is based on the average and standard deviation (SD) of the full season for each parameter.

In order to consider the influence of match location on results, the "Britannic Ranking" was used to determine positive, negative and neutral performance. More precisely, a bonified victory (offensive bonus) during a home match, a defensive bonus or a victory during an away match will be considered as "positive performance". Defeat during a home match will be considered as "negative performance", and finally, a victory during a home match and a defeat during an away match will be considered as "neutral performance". This type of ranking is often used by French rugby teams' staff to predict final standings when considering the number of remaining matches to be played at home and/or away.

### Data contextualization and transformation:

As demonstrated in other studies, 2WL and match performance are influenced by different contextual factors such as: the period of the season, player status (starter, substitute), playing position and match location, among other factors [6,21]. Therefore, the player's status (starter or substitute) and position (forward or back) on the field was taken in account for this study.

To study the effects at short and moderate terms, all WL parameters were analyzed on a rolling average. The rolling average for the 2nd, 3rd, 4th, 5th, 6th, 7th and 8th previous weeks was analyzed for each parameter. A weighted average, to increase the impact of recent WL, was also used with similar time lags. Variability of training was considered by analyzing the SD of previous weeks (2nd to 8th). Finally, for each WL parameter (Table 2), 21 other parameters were added: 7 for average at 7 different weekly considerations, 7 for weighted average and 7 for SD.

After one year of data collection, an important analysis was performed in an attempt to analyze how WL influences match performance in successive matches for an elite RU team. This study provides a methodology based on data mining to relate physical performance variations of players during time-framed training sessions and their performance throughout the following matches. The study is structured by three major steps, each one being associated to different analysis methodologies. The first part focuses on constructing an informative dataset from GPS measurements and specific data on WL, match activity and performance indicators. The second part analyses this information to identify rugby specific actions in terms of player status (starter or substitute). The third part aims at identifying links between performance and match activity. Difficulties were encountered on the two previous parts. Indeed, it was necessary to identify useful information from such large amounts of dataset as to optimally interpret the data.

### Statistical analysis

As shown in Fig 1, the methodological process can be divided into three steps. As a preliminary step, descriptive statistics were computed. Prior to the main analysis, the level and the variability (mean ± SD) of each training parameter were calculated relatively to playing position and the player's status using a linear mixed model. Effect size (ES) was then calculated using Cohen's d statistics where an ES <0.2 was considered non-significant (NS), 0.2–0.6 small, 0.6–1.2 moderate, 1.2–2.0 large and > 2.0 very large [22,23].

Because performance is measured through a set of several variables and not a unique response variable, multivariate statistical approaches were carried out. With the same constraints and objectives, Haghighat et al. [24] propose a review of several methods to allow an automatic selection of the most significant features based on data mining techniques. We favor a dimensional reduction approach as it facilitates analysis during the third part, makes storage/computation less expensive and allows for easier interpretation [17]. For this purpose, a linear dimension reduction method called Principal Component Analysis (PCA) was used on the performance dataset to reduce the dimension of analysis [17]. A normalized PCA was used in the second part to reduce the high-dimensional raw feature [17,25]. PCA is a descriptive

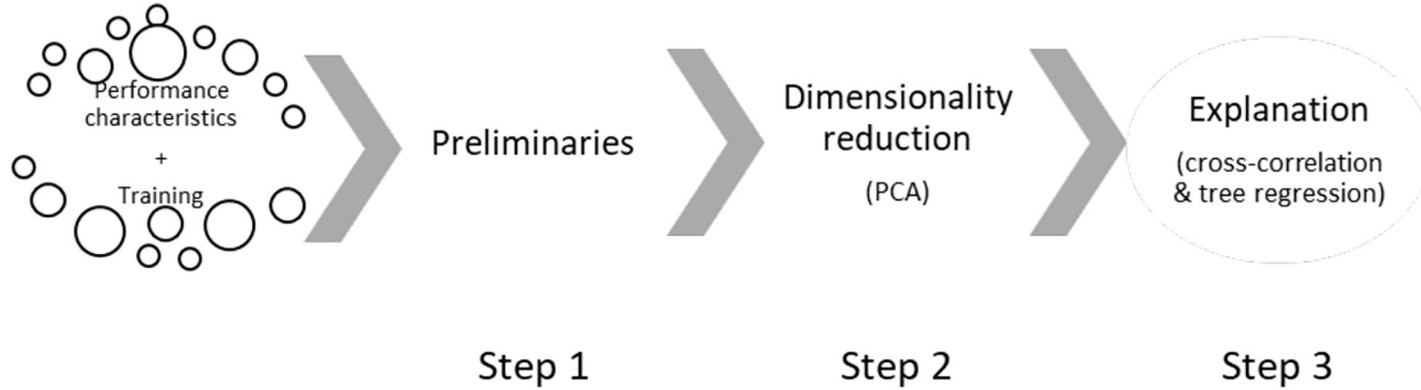

*PCA: Principal component analysis*

**Fig 1. Multivariable statistical approach.**

multivariate statistical analysis that explores a set of quantitative variables in order to improve collinearity between them and to discuss the importance of each variable in terms of variability. It is a mathematical tool used for computing a set of new synthetic variables. These variables, also called dimensions, aim at identifying high variability components based in bigger dimensional datasets. Subsequently, choosing a small number of new dimensions allows to create a discriminative sub-space based on informative features in terms of variability to map the high-dimensional data set.

Finally, in the third step, we tried to explain the relationships between different WL parameters at short and moderate terms (x-factors) and the performance/locomotor activity indicators (y-factors). A cross-correlation analysis was used to assess the level of cross-collinearity between performance descriptors (response variables) and training locomotor activity descriptors (descriptive variables). However, there exists a certain limit for linear models to highlight relationships between WL and performance (no significant correlation was found between the two groups). Regression trees are also applied during this third step to extract discriminative information for performance and (potentially) to further reduce the dimension. A regression tree is a data mining process which is based on decision induction analysis. It estimates a regressive relationship through binary partitioning (splitting) by testing the link between a set of explanatory variables and a quantitative response variable. Classical and conditional regression trees were used to identify non-linear link through a graphical binary tree. This results in a discrete model based on a set of rules given by a categorical pattern of dependence computed on interaction between categorical explanatory variable and categorized quantitative explanatory ones. In this part, the different response variables used are successively the first two dimensions of the PCA above. All analysis was conducted with R Statistical Software (R. 3.3.3, R Foundation for Statistical Computing).

## Results

Workload description, collective performance and specific indicators of team performance:
Table 3 shows the 2WL of players depending on their position and their playing status. Regarding the playing position, our results show no significant difference concerning the

Table 3. Weekly workload parameters depending on playing position and players status during matches.

| Playing position and players status groups | S-RPE (AU) | Volume (h) | Strain (AU) | TD (m) | HSR distance (m) | HMP distance (m) | TRIMPS (AU) | TS > 85% HR$_{max}$ (min) | Heavy impacts (n) | NBL (AU) |
|---|---|---|---|---|---|---|---|---|---|---|
| Forwards (n = 116) | 2369.6 ± 475.3 | 8.0 ± 1.5 | 2418.9 ± 697.7 | 11481.5 ± 3238.7 | 1148.1 ± 468.4 | 2713.4 ± 1871.6 | 807.5 ± 434.9 | 22.9 ± 13.2 | 32.8 ± 36.0 | 219.0 ± 70.6 |
| Backs (n = 164) | 2277.5 ± 525.9 | 7.8 ± 1.6 | 2270.8 ± 776.5 | 14382.2 ± 4041.8*** | 2515.6 ± 785.2*** | 3620.7 ± 2049.5*** | 834.5 ± 382.3 | 26.4 ± 15.7 | 37.2 ± 51.0 | 320.6 ± 162.2*** |
| E.S. | NS | NS | NS | 0.8 | 1.6 | 0.5 | NS | NS | NS | 0.8 |
| Starters (n = 244) | 2276.5 ± 361.6 | 7.6 ± 1.3 | 2256.5 ± 657.6 | 12534.7 ± 3316.8 | 2011.8 ± 796.1 | 2574.6 ± 860.6 | 764.0 ± 329.0 | 22.9 ± 12.6 | 26.5 ± 32.7 | 262.0 ± 106.9 |
| Substitutes (n = 84) | 1660.2 ± 466.7*** | 7.0 ± 1.8 | 1158.3 ± 443.2*** | 11769.3 ± 3168.3 | 1613.5 ± 665.0*** | 2189.4 ± 761.3*** | 681.0 ± 301.0* | 21.3 ± 14.0 | 23.2 ± 32.5 | 231.6 ± 115.5 |
| E.S. | 1.4 | 0.5 | 1.1 | NS | 0.5 | 0.5 | 0.3 | NS | NS | NS |

S-RPE: Session of rating perceived exertion; TD: Total distance; HSR: High-speed running; HMP: High-metabolic power; TRIMPS: Training impulses; TS: Time spent; HR$_{max}$: Maximal heart rate; NBL: New body load; E.S.: Effect size.

* $p<0.05$

*** $p<0.001$; significant differences between forwards and backs or between starters and substitutes

internal WL when the S-RPE method was used. However, backs covered a greater TD ($p<0.001$, $d = 0.8$) and have higher NBL ($p<0.001$; $d = 0.8$) than forwards. This is more pronounced in faster speed ($p<0.001$, $d = 1.6$) and metabolic zone ($p<0.001$, $d = 0.5$). No other significant differences were observed between backs and forwards concerning the other 2WL parameters. Table 3 also highlighted that players who started the upcoming match were exposed to greater WL regardless of their position. This was true for the weekly S-RPE ($p<0.001$, $d = 1.4$), HSR and HMP distances ($p<0.001$, $d = 0.5$, respectively) and TRIMPS ($p<0.05$, $d = 0.3$).

Table 4 provides information about the 2WL parameters, at short and moderate terms, depending on the team's performance (victory vs defeat or positive vs negative) during official matches. It shows that, when the team studied won, some 2WL parameters were greater during the week prior to competition. Indeed, the acute S-RPE was greater ($p<0.001$, $d = 0.4$), as well

Table 4. Weekly workload parameters depending on team performance during matches.

| Collective performance | Acute S-RPE (AU) | Chronic S-RPE (AU) | Acute:chronic S-RPE (AU) | Acute TD (m) | Chronic TD (m) | Acute:chronic TD (AU) | Acute TRIMPS (AU) | Chronic TRIMPS (AU) | Acute H.I. (n) | Chronic H.I. (n) |
|---|---|---|---|---|---|---|---|---|---|---|
| Victory (n = 97) | 2396.4 ± 404.5 | 2043.7 ± 356.9 | 1.19 ± 0.2 | 13066.4 ± 3095.7 | 10824.5 ± 3139.9 | 1.29 ± 0.6 | 769.0 ± 258.0 | 720.7 ± 273.7 | 24.6 ± 29.5 | 23.9 ± 28.3 |
| Defeat (n = 147) | 2197.3 ± 489.0*** | 1980.7 ± 348.3 | 1.12 ± 0.2* | 12191.2 ± 3418.6* | 10898.9 ± 2663.8 | 1.14 ± 0.4 | 760.7 ± 368.4 | 704.6 ± 224.1 | 27.7 ± 34.7 | 29.1 ± 30.2 |
| E.S. | 0.4 | NS | 0.3 | 0.3 | NS | NS | NS | NS | NS | NS |
| Positive (n = 53) | 2324.2 ± 384.1 | 2037.4 ± 364.6 | 1.16 ± 0.2 | 12507.1 ± 2408.4 | 10766.6 ± 3193.2 | 1.22 ± 0.3 | 804.5 ± 289.2 | 781.3 ± 304.6 | 20.4 ± 20.3 | 25.7 ± 29.7 |
| Negative (n = 26) | 2294.7 ± 441.3 | 2165.7 ± 366.0 | 1.07 ± 0.2 | 11518.2 ± 2471.5 | 11848.1 ± 3131.2 | 1.04 ± 0.3 | 768.8 ± 342.0 | 741.4 ± 228.9 | 42.2 ± 26.6*** | 50.1 ± 31.1*** |
| E.S. | NS | NS | NS | NS | NS | NS | NS | NS | 1.0 | 0.8 |

S-RPE: Session of rating perceived exertion; TD: Total distance; TRIMPS: Training impulses; H.I.: Heavy impacts; NBL: New body load; E.S.: Effect size

* $p<0.05$

** $p<0.01$

*** $p<0.001$; significant differences between victory and defeat or between positive or negative result.

**Table 5. Individual indicators of match's performance depending on the positions and match final results.**

| Playing position | Collective performance | Avg. Speed (m.min⁻¹) | Relative HSR (m.min⁻¹) | Relative VHSR (m.min⁻¹) | Relative HMPD (m.min⁻¹) | Sprints (n) | H.I (n) | ST (n) | OT (n) | ODW (n) | Penalty Conc. (n) | Ruck (n) | LB (n) | Ball hit (n) | Activity index |
|---|---|---|---|---|---|---|---|---|---|---|---|---|---|---|---|
| Forwards | Victory (n = 43) | 57.4 ± 7.1 | 21.6 ± 9.4 | 5.3 ± 5.8 | 32.8 ± 9.5 | 0.5 ± 1.1 | 31.5 ± 30.8 | 10.1 ± 3.7 | 2.1 ± 1.2 | 1.1 ± 2.4 | 0.73 ± 7.1 | 17.4 ± 5.4 | 0.17 ± 0.8 | 9.2 ± 9.4 | 39.9 ± 13.4 |
|  | Defeat (n = 58) | 58.6 ± 6.6 | 23.8 ± 8.8 | 6.2 ± 5.5 | 41.1 ± 9.1 | 0.4 ± 0.8 | 30.1 ± 27.4 | 7.3 ± 3.4* | 1.3 ± 1.1* | 1.9 ± 2.1 | 0.61 ± 0.9 | 16.3 ± 5.0 | 0.17 ± 0.7 | 6.7 ± 8.8 | 33.7 ± 12.6* |
|  | E.S. | NS | NS | NS | NS | NS | NS | 0.8 | 0.6 | NS | NS | NS | NS | NS | 0.5 |
| Backs | Victory (n = 52) | 67.6 ± 6.4$ | 33.2 ± 8.6$ | 13.8 ± 5.3$ | 35.3 ± 8.4$ | 4.1 ± 3.2$ | 24.5 ± 29.6 | 5.3 ± 3.3$ | 1.1 ± 1.0$ | 1.5 ± 2.2 | 0.18 ± 0.86$ | 5.3 ± 4.9$ | 0.39 ± 0.7 | 10.3 ± 8.6$ | 24.6 ± 12.2$ |
|  | Defeat (n = 88) | 65.3 ± 6.2*,$ | 35.7 ± 8.4$ | 15.5 ± 5.2$ | 44.2 ± 8.4*,$ | 4.9 ± 3.1$ | 32.7 ± 27.4 | 4.8 ± 3.3$ | 0.8 ± 1.0$ | 1.9 ± 2.1 | 0.31 ± 0.83$ | 5.6 ± 4.8$ | 0.47 ± 0.7$ | 13.6 ± 8.4*,$ | 27.6 ± 11.9$ |
|  | E.S. | 0.4 | NS | NS | 1.0 | NS | NS | NS | NS | NS | NS | NS | NS | 0.4 | NS |
| Position E.S. |  | 1.2 | 1.4 | 1.7 | 0.4 | 1.3 | NS | 1.0 | 0.6 | NS | 0.5 | 2.3 | 0.4 | 0.5 | 0.8 |
| Forwards | Positive (n = 18) | 57.1 ± 5.3 | 23.9 ± 6.2 | 6.2 ± 5.1 | 35.5 ± 6.4 | 0.2 ± 0.8 | 30.4 ± 28.4 | 8.4 ± 2.1 | 2.1 ± 1.1 | 1.4 ± 2.0 | 0.56 ± 0.6 | 19.5 ± 4.8 | 0.40 ± 0.8 | 10.6 ± 9.4 | 42.6 ± 11.7 |
|  | Negative (n = 10) | 58.5 ± 5.8 | 23.1 ± 10.2 | 5.8 ± 4.7 | 34.6 ± 10.5' | 0.2 ± 0.9 | 42.8 ± 31.1 | 5.2 ± 3.5* | 1.3 ± 1.0* | 3.2 ± 2.1* | 0.90 ± 0.6* | 16.7 ± 5.3 | 0.10 ± 0.9 | 8.3 ± 9.1 | 32.8 ± 12.8* |
|  | E.S. | NS | NS | NS | NS | NS | NS | 1.1 | 0.6 | 0.9 | 0.6 | NS | NS | NS | 0.8 |
| Backs | Positive (n = 27) | 66.5 ± 5.3$ | 33.9 ± 5.5$ | 20.3 ± 4.9$ | 42.6 ± 5.6$ | 4.2 ± 2.6$ | 24.5 ± 28.4 | 4.0 ± 1.9$ | 1.2 ± 1.1$ | 1.9 ± 1.3 | 0.04 ± 0.6$ | 4.9 ± 4.8$ | 0.56 ± 0.8 | 14.4 ± 8.3 | 27.2 ± 11.7$ |
|  | Negative (n = 14) | 69.9 ± 5.5*,$ | 44.7 ± 10.9*,$ | 13.8 ± 4.7$ | 55.8 ± 11.2*,$ | 5.7 ± 2.7$ | 55.7 ± 29.4* | 4.1 ± 3.7 | 0.9 ± 1.1 | 3.9 ± 2.0* | 0.16 ± 0.6$ | 6.0 ± 5.0$ | 0.54 ± 0.8 | 14.3 ± 8.7$ | 29.5 ± 10.2 |
|  | E.S. | 0.5 | 1.2 | 1.4 | 1.2 | NS | 0.4 | NS | NS | 1.0 | NS | NS | NS | 0.4 | NS |

Avg. speed: Average speed; HSR: High-speed running; VHSR: Very high-speed running; HMPD: High-metabolic power distance; H.I.: Heavy impacts; ST: Successful tackles; OT: Offensive tackles; ODW: Offensive duel won; Penalty conc.: Penalty conceded; LB: Defensive line breaks. E.S.: Effect size.

* p<0.05; significant differences between victory-defeat and positive and negative results from Britannic Ranking.

$ p<0.05; significant differences between forwards and backs.

as acute:chronic S-RPE *(p<0.05, d = 0.3)* and acute TD *(p<0.05 d = 0.3)*. When using the British ranking system to analyze collective performance, other significant differences were observed concerning the influence of 2WL parameters on collective performance. In particular, acute HI *(p<0.001, d = 1.0)* and chronic HI *(p<0.001, d = 0.8)* were significantly greater during the weeks with negative results (defeat at home).

Table 5 highlights the individual indicators of match performance according to the player's position and match results. It reveals that backs have a greater average speed (m.min⁻¹) during matches won *(p<0.05, d = 0.4)*. In contrast, the relative distance traveled in HMP zone is significantly greater in backs *(p<0.5, d = 1.0)* during lost matches. Concerning specific activities, forwards performed a bigger defensive performance during successful matches by totalizing more completed tackles *(p<0.05, d = 0.8)* and more offensive tackles *(p<0.05, d = 0.6)* compared to matches that were lost. Moreover, forwards have a greater activity index during successful matches *(p<0.05, d = 0.5)*. On the contrary, backs played the ball significantly less during victories *(p<0.05, d = 0.4)*. Table 5 also highlights other significant differences between backs and forwards concerning physical and rugby specific actions during matches.

Table 6 shows the influence of player status on playing activity during matches. Playing activity indexes were greater for substitutes independently of the player's position *(p<0.05, d = 1.0 & d = 1.9, respectively for forwards and backs)*. Furthermore, forward substitutes conceded less penalties (relatively to ball-in-play time—when the player played) compared to starting forwards *(p<0.05, d = 0.9)*.

## Summary of individual performance:

Characteristics of individual speed are used in this analysis. Fig 2 shows more important collinearity for three variables (HMPD.min, HSR.min and TD.min) with Dim1 and that only sprint

**Table 6. Individual indicators of match's performance depending on the player's status (starters vs substitutes).**

| Playing position | Collective performance | Avg. Speed (m. min⁻¹) | Relative HSR (m.min⁻¹) | Relative VHSR (m. min⁻¹) | Relative HMPD (m. min⁻¹) | Sprints (m. min⁻¹) X 10 | H.I (n. min⁻¹) | ST (n. min⁻¹) X 10 | OT (n. min⁻¹) X 10 | ODW (n. min⁻¹) X 10 | Penalty Conc. (n. min⁻¹) X 10 | Ruck (n. min⁻¹) X 10 | LB (n. min⁻¹) X 10 | Ball hit (n. min⁻¹) X 10 | Relative Activity index |
|---|---|---|---|---|---|---|---|---|---|---|---|---|---|---|---|
| Forwards | Starters (n = 78) | 58.3 ± 8.8 | 23.1 ± 9.8 | 5.8 ± 5.9 | 34.4 ± 9.4 | 0.16 ± 0.1 | 1.1 ± 0.9 | 3.1 ± 1.3 | 0.59 ± 0.38 | 0.59 ± 0.73 | 0.25 ± 0.29 | 6.1 ± 1.8 | 0.06 ± 0.09 | 2.8 ± 1.5 | 0.47 ± 0.18 |
| | Substitutes (n = 16) | 59.0 ± 8.8 | 25.0 ± 9.8 | 6.6 ± 5.9 | 35.9 ± 9.4' | 0.33 ± 0.1 | 1.2 ± 0.9 | 3.6 ± 1.4* | 0.37 ± 0.38*$ | 0.23 ± 1.0 | 0.08 ± 0.02* | 6.0 ± 1.7 | 0.16 ± 0.10 | 3.3 ± 1.6 | 0.85 ± 0.25* |
| | E.S. | NS | NS | NS | NS | NS | NS | NS | 0.5 | NS | 0.9 | NS | NS | NS | 1.0 |
| Backs | Starters (n = 121) | 66.1 ± 8.7$ | 34.8 ± 9.7$ | 14.9 ± 5.9$ | 43.1 ± 9.6$ | 1.55 ± 1.0$ | 1.0 ± 0.9 | 1.7 ± 1.2$ | 0.32 ± 0.35$ | 0.60 ± 0.72 | 0.09 ± 0.03$ | 1.9 ± 1.8$ | 0.06 ± 0.8$ | 4.4 ± 1.5$ | 0.45 ± 0.17 |
| | Substitutes (n = 26) | 65.7 ± 8.6·$ | 37.2 ± 9.6$ | 15.3 ± 5.8$ | 43.1 ± 9.3$ | 5.7 ± 2.7$ | 1.4 ± 0.9 | 2.0 ± 1.3 $ | 0.10 ± 0.3*$ | 0.63 ± 0.71 | 0.10 ± 0.05 | 1.8 ± 1.8$ | 0.34 ± 0.11*$ | 6.2 ± 1.5*$ | 1.14 ± 0.30*$ |
| | E.S. | NS | NS | NS | NS | NS | NS | NS | 0.6 | NS | NS | NS | 0.4 | 0.6 | 1.9 |
| Position E.S. | | 0.9 | 1.2 | 1.7 | 0.9 | 1.9 | NS | 1.1 | 0.7 | NS | 0.4 | 2.4 | 0.4 | 0.6 | 0.5 |

Avg. speed: Average speed; HSR: High-speed running; VHSR: Very high-speed running; HMPD: High-metabolic power distance; H.I.: Heavy impacts; ST: Successful tackles; OT: Offensive tackles; ODW: Offensive duel won; Penalty conc.: Penalty conceded; LB: Defensive line breaks. E.S.: Effect size.

* p<0.05; significant differences between starters-substitutes.

$ p<0.05; significant differences between forwards and backs.

and accelerations (Sp+Acc) are highly correlated with Dim2. One can consider that the three variables of Dim1 measure the same aspect of performance, while Sp+acc measure another aspect which is not correlated with the others. With these two new synthetic dimensions, around 88% of variability for the measures can be explained. The first dimension (Dim1) contained 65.43%, while 23.22% was explained by the 2nd dimension (Fig 2). The heterogeneity between the observations is meanly due to the variables contained in Dim1 called "running. performance" and can be interpreted as follow: a negative value means "low performance" and a high positive value means "high performance".

Ten characteristics of match playing activities are used in this analysis but only the meaningful ones can explain the variability of observations (cos2>0.5). They are shown on Fig 3. A larger degree of collinearity was seen between Tack and Tack.suc than in activity rate and Ms. win. Moreover, these 2 groups show no correlation. However, the PCA is not very efficient here because only 38.51% of total variability is explained through these 2 dimensions. The first dimension (Dim1) contained 21.6%, while 16.91% was measured by the 2nd dimension. It outlines the fact that the 10 characteristics have no signs of correlation between each other. Other links may exist but these are not detectable by linear methods such as PCA.

## Performance insights from descriptors of training activity

As a preliminary analysis, several correlational matrices were calculated to assess the level of collinearity between WL indicators (explanatory variables gathered into a matrix called X), performance indicators (variables to be explained gathered into a matrix called Y) and the cross-correlation between X and Y. The results are presented using a black and grey colored gradient where dark colors represent strong correlations, positive or negative (Fig 4). No significant collinearity is noted in the cross-correlational matrix thus encouraging the use of nonlinear statistics analytical tools to study potential links between WL and performance indicators.

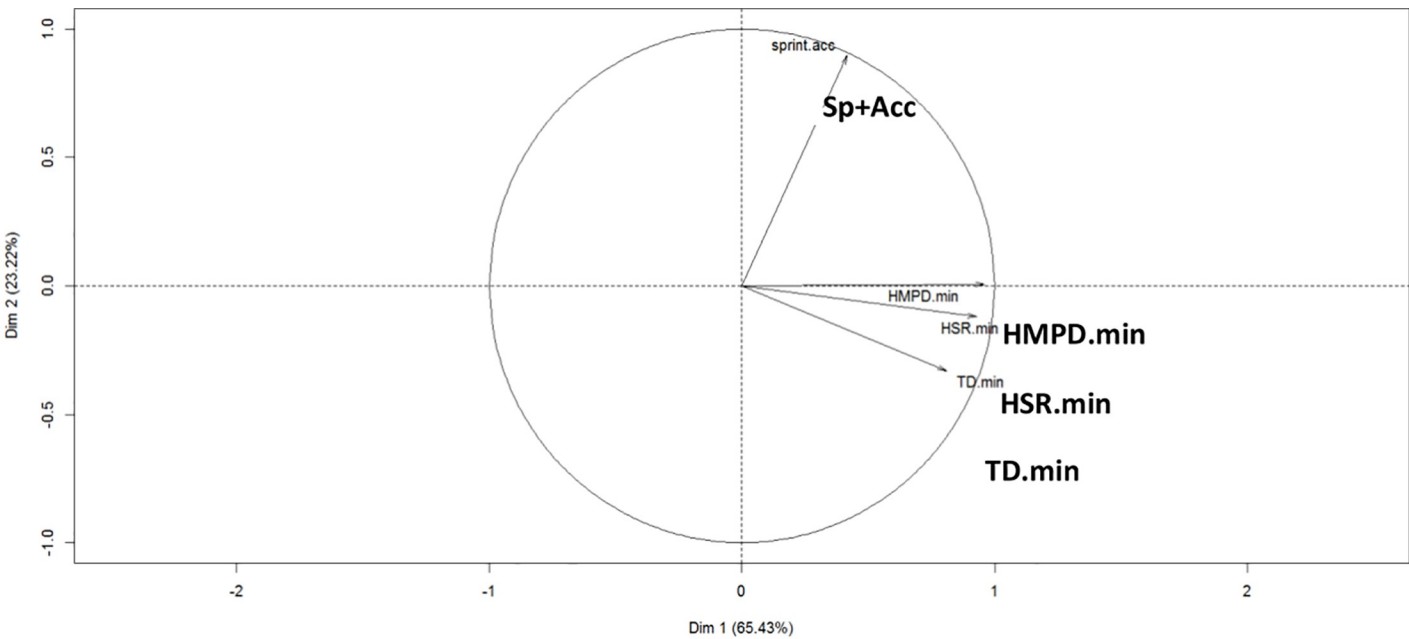

*Sp+Acc: Number of sprints and accelerations; HMPD.min: High-metabolic power distance divided by minutes of ball-in-play time.; HSR.min: High-speed running distance divided by minutes of ball-in-play time.; TD.min: Total distance divided by minutes of ball-in-play time; Dim 1: Dimension 1; Dim 2: Dimension 2*

**Fig 2. First principal plane from PCA on normalized (Z-score) individual speed descriptors.**

"Running.performance", the first principal component of PCA corresponding to a performance descriptor, is analyzed hereby (Fig 5). On the left-hand side of the tree, node number 2 characterizes the mean level of performance of backs which is logically lower than forwards in terms of offensive activity. Node 19 contains observations on backs with greater levels of offensive performance. This is due to an acute amount of speed exertion > 21e+3 combined with a 4-week rolling average of heavy impacts (Hi.4weeks.SD) < 9.4 heavy impacts. Node 8 contains observations on backs with lower levels of offensive activity which is due to a Hi.4weeks.SD >= 9.4. According to the right branch of the tree, the higher level of offensive performance for forwards is due to a 6-week average of low running speed (LSR.moy.6) <= 8421. This regression tree illustrated that several parameters appear to influence running performance but have no significant effect according to a statistical test (conditional regression tree). Concerning the other significant effects observed on the relationships between WL parameters and activity indicators, Fig 6 reveals that the normalized (Z-score) number of sprints and accelerations was significantly and negatively affected when the average time spent under 85% $HR_{max}$ was above 218.992min (Fig 6A). Similarly, the normalized number of offensive duels won was significantly and positively affected by the chronic (4 week rolling average) number of HI (Fig 6B).

Table 7 presents the summary of all the conditions tested when using this method.

## Discussion

The main goal of this study was to detect the existing relationships between WL at short and moderate terms and performance or locomotor activity during matches of professional RU players throughout a season. Several preliminary steps were necessary to accomplish this: i)

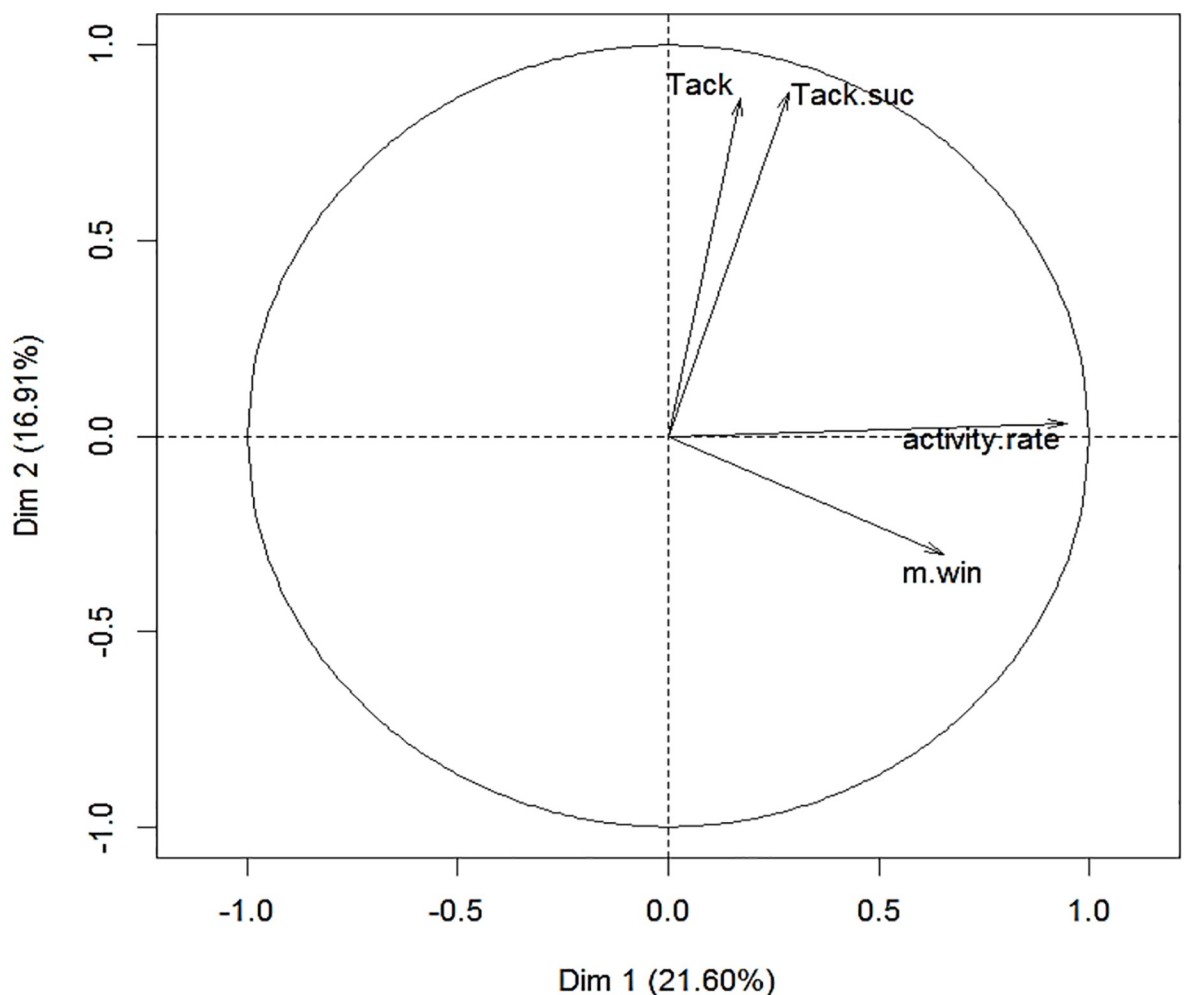

*Tack: Number of tackled attempted; Tack.suc: Number of successful/completed tackle; Ms.won: Number of meters won by carrying the ball; Dim 1: Dimension 1; Dim 2: Dimension 2.*

**Fig 3. First principal plane from PCA on specific activities.**

understanding the different factors that might influence WL and team performance, ii) synthesizing activity indicators to facilitate and simplify the modelling/definition of individual performance and iii) analyzing, with different statistical models (linear and threshold-based), the influence of different WL parameters on performance. The main findings showed that 2WL was influenced by playing position and player status. Indeed, backs presented greater external WL (GPS-based data) during training sessions than forwards (p<0.001), while starters expressed greater internal (S-RPE and TRIMPS methods) and external WL (p<0.001). Furthermore, when team performance was analyzed considering home advantage (using Britannic ranking classification), the findings demonstrated that collision load (number of heavy impacts) at short and moderate terms negatively influenced team performance (p<0.001). Positive team performance was noticed when backs covered greater relative distance (m.min$^{-1}$) and when forwards successfully tackled more. Furthermore, forwards also had a greater (p<0.05) activity index during victory. Concerning our attempt to define performance with new synthetic variables, we found that one can construct a synthetic index based on individual

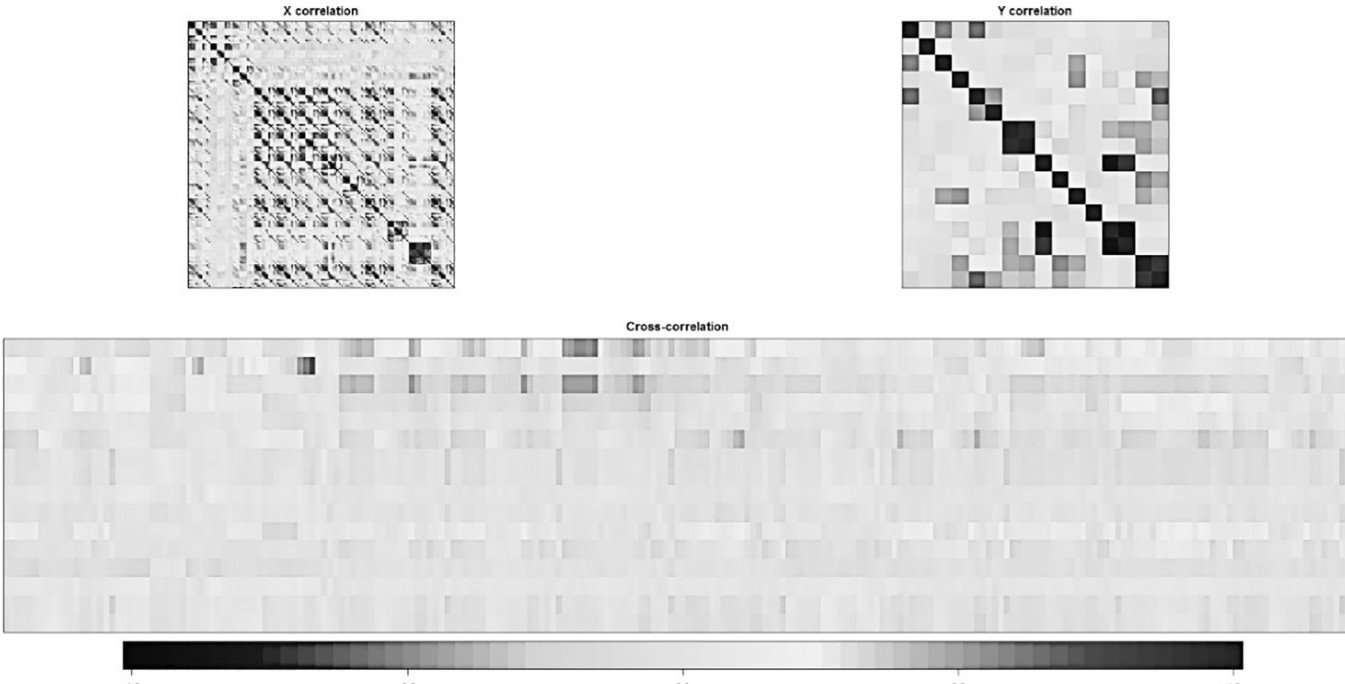

The intensity of the color shows the level of correlation between the variables. X variables represented all the workload data collected and time-declined, while Y variables represented all the match performances data relativized to the ball-in-play time.

**Fig 4. Correlation matrices between the variables "X" of the weekly workload and the parameters "Y" of the game's activity.**

GPS data (individual Z-score, based on the average and SD for each player and for each GPS parameters throughout the season) regrouping the HMPD.min, HSR.min, TD.min, on one side, and Sp+Acc on the other. No synthetic index was found concerning specific activity parameters. Finally, the first method conducted to highlight the relationships between 2WL and individual locomotor activity/performance indicators, based on a linear model, did not allow for any observation of significant effect. The use of a threshold model, from data mining processes, permitted us to illustrate some significant effects of WL parameters on individual performance indicators. Indeed, the chronic (4-week rolling average) number of heavy impacts influenced positively the number of duels won during matches (p<0.05). Finally, these results highlight the difficulty to identify and synthesize physical performance in RU and also point out the high level of complexity encountered when establishing models to establish relationships between WL and physical performance/locomotor activity during matches.

Before analyzing the influence of WL on performance, it was important to underline the different factors which influence 2WL. Indeed, several studies reported significant differences between forwards and backs concerning the internal and external WL of training weeks during preseason and in-season for professional rugby players [6,26]. The present findings confirmed, in part, the results arguing that backs have greater external WL than forwards. These differences were mainly explained by the difference in locomotor activity during training sessions. Nevertheless, no significant differences were observed regarding the internal WL indicators (S-RPE & TRIMPS). The scrum and lineout training sessions for forwards, which account for

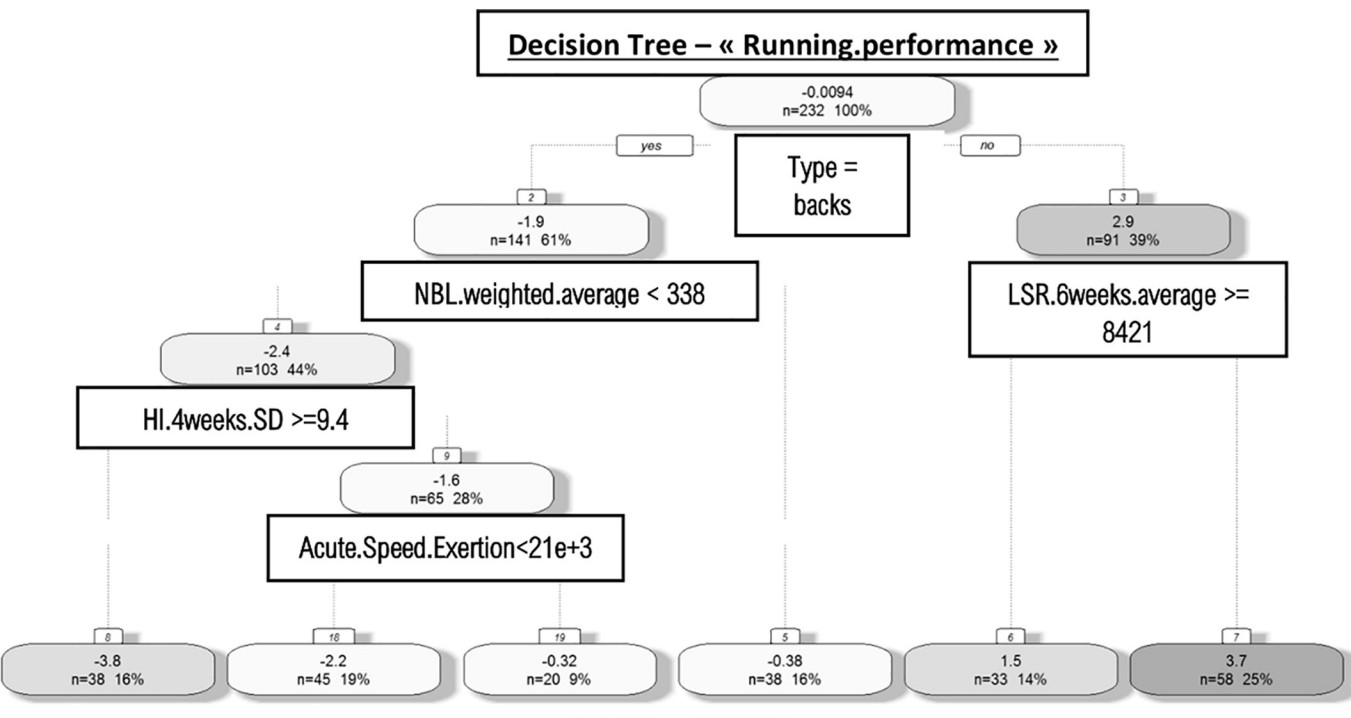

*n denotes the number of observations in each node, boxplot in each terminal node shows the level and the variability of "Running.performance".*

**Fig 5. Traditional decision tree illustrating the level of "Running performance".**

20 to 30% of weekly training, do not generate running activity. Thus, external WL during this type of training cannot be recorded by GPS technology. Therefore, the differences observed in external WL between forwards and backs, do not reflect the difference in quantity of training. It is also important to note that the external load estimate by GPS technology may presents some limitations about the accuracy and the reliability of some variables like the accelerations and the metabolic power approach variables [27]. These limitations lead us to remain cautious about the indiscriminate analysis of data derived from GPS signals [27]. Playing position was not the only factor that explained the observed differences among WL parameters. In a similar way to a study carried out in soccer [21], we have analyzed the relationships between players' status and WL. Our findings demonstrate that substitute players, regardless of their position, had lower internal and external WL than starting players during weekly training. These results must be considered in moderate and long-term training processes. Indeed, players who substitute regularly were exposed to a lower internal and external WL. This trend may conduct to undertraining. Thus, team staffs should propose complementary training to these players to expose them to high intensity running and HR efforts. In their recent study, Dalton-Barron et al. [28] demonstrated that WL perception was influenced by different factors: playing position, previous match results, phase of the season and the time lapse between matches. Therefore, results from this study and prior ones, point out to the need for applying a multifactorial approach to plan and monitor the rugby players' WL during the different phases of the season (Tables 3 & 4).

Dalton-Barron's [28] study also reflects the significant impact of the competitive context on the perception of difficulty of training sessions during a "competitive-phase" week. Indeed,

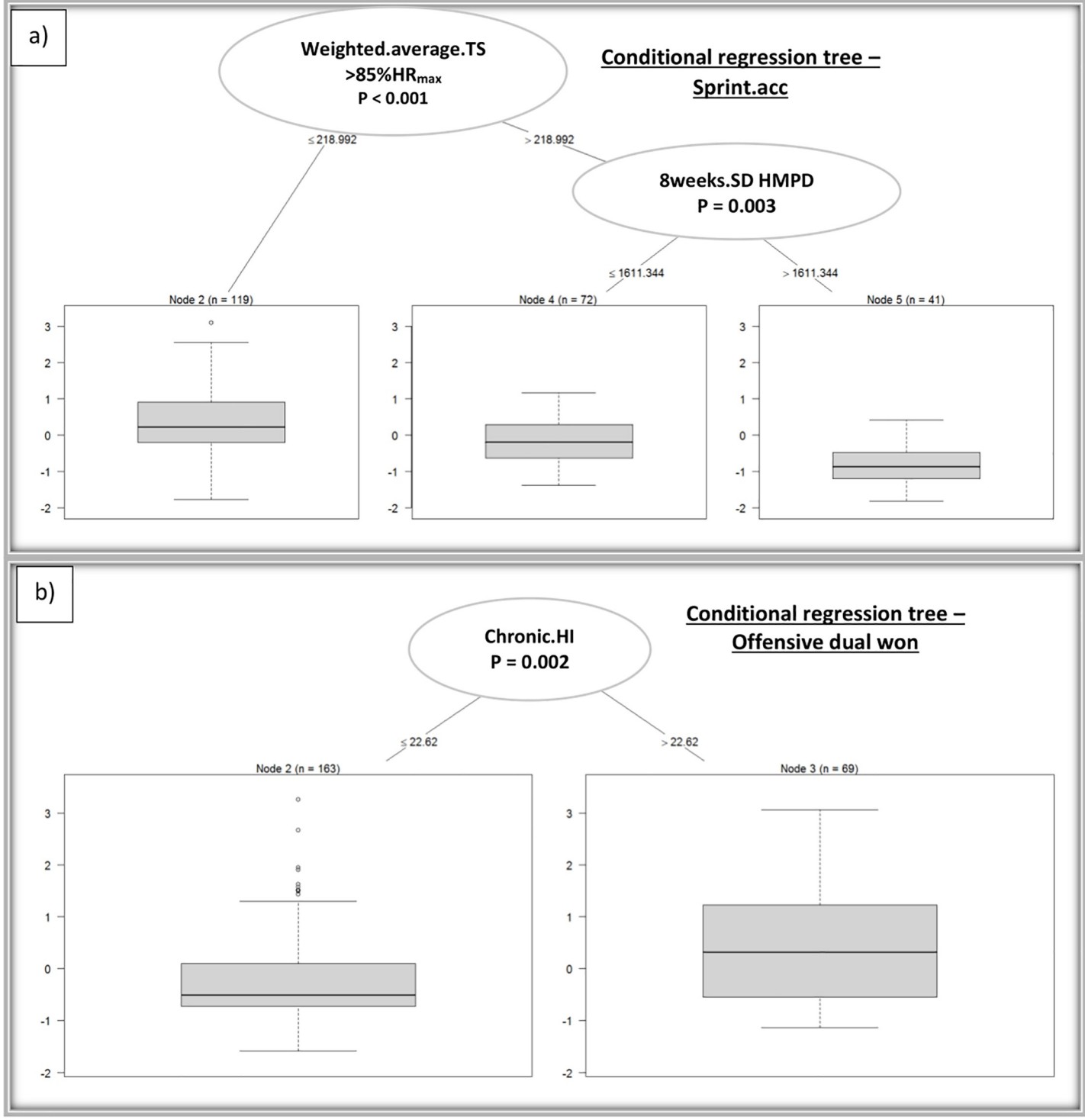

*Sprint.acc: Number of sprints and accelerations during the matches; Weighted.average.TS>85%: Weighted average for the 4 last weeks of the time spent under 85% of maximal heart rate; 8weeks.SD.HMPD: Standard deviation of the 8 last weeks about the high-metabolic power distances; Chronic HI: chronic number (average of 4 last weeks) of heavy impacts.*

**Fig 6.** Conditional regression trees showing the influence of workload parameters on some activity indicators during the matches: a) number of sprints and accelerations, b) number of offensive duals won.

**Table 7. Overview of the different analysis perform to observe if some workload indicators influence (positively or negatively) the different performance/locomotor activity indicators.**

| TYPE OF INDICATORS | PERFORMANCE PARAMETERS | POSITIVE INFLUENCING FACTOR(S) | NEGATIVE INFLUENCING FACTOR(S) |
|---|---|---|---|
| SYNTHETIC INDEX OF RUNNING PERFORMANCE | « Running.Performance » | / | / |
| | Sprint.Acc | Weighted average of time spent under 85% of $HR_{max}$ $\leq$ 219 min. | Weighted average of time spent under 85% of $HR_{max}$ > 219 min. |
| SYNTHETIC INDEX OF SPECIFIC ACTIVITY | Defensive performance | Forwards | Backs 8 weeks average for number of severe impacts <19.62 impacts. |
| INDIVIDUAL NORMALIZED (Z-SCORE) PERFORMANCE INDICATORS | Tackle attempted | / | / |
| | Tackle succeed | / | / |
| | % Tackle succeed | / | / |
| | Offensive tackle | / | / |
| | Offensive duals win | Average of HI for the 4th last weeks > 22.62 impacts. | Average of HI for the 4th last weeks $\leq$ 22.62 impacts. |
| | Ruck participation | / | / |
| | Meters win | / | / |
| | Number of balls played | / | / |
| | Line break | / | / |
| | Penalty conceded | / | / |
| | Activity index | / | / |

$HR_{max}$: maximal heart rate; HI: heavy impacts.

these results corroborate other studies which also demonstrated an effect of competitive stress on physiological adaptations, especially on endocrine responses [6,8,29]. In our findings, significant differences were found when comparing WL parameters during weeks with a match victory (Table 4). The differences were mainly observed in the subjective perception (S-RPE) method and may suggest that stress before key matches of the season could induce an increase in WL perception. Indeed, in RU, home advantage was statistically demonstrated [30,31]. The team studied here was a bottom ranked team for which home matches were of crucial importance. Indeed, during season studied, the team won 14 of their 16 home matches and only won one away game (out of 16). Thus, the team studied prepared home matches with particularly high pressure. Subjective perception of difficulty of training sessions seems to be influenced by the competitive context ($p < 0.001$, d = 0.4). Thus, a greater acute total distance ($p < 0.05$, d = 0.3) was also observed and may signal a greater WL exposition during the weeks of victory. Trainers seemed to have a tendency (maybe unconsciously) to increase the external WL by increasing tactical and strategical situations to prepare for a challenging match. In an attempt to minimize the influence of match location on team performance (especially with the bottom ranking team), the Britannic ranking classification was applied to highlight the "positive" results (defensive bonus, draw and victory during away matches and bonified victories for home and away venues). With this filter of team performance, it appears that an important number of heavy impacts at short and moderate terms influenced performance negatively. This result could suggest that neuromuscular fatigue induced by repetitive heavy impacts [32,33] may also affect team performance during matches. However, because of our relatively small dataset, great caution should be taken when analyzing these results. Moreover, it is crucial to specify that all these results depend on multiple contextual factors (score, domiciliation, level of the opposition, climatic conditions, . . .) [34]. Indeed, studying one team during a

unique season represents a complex protocol. Therefore, the results observed are highly linked to the context and specifity of the team.

In terms of performance, it also seems that different individual locomotor activity parameters influence team performance (Table 5). Thus, the specific activity (number of actions normalized by ball-in-play time) and defensive performance of forwards (number of completed and offensive tackles) were greater during matches won (p<0.05). Thus, the number and percentage of completed tackles seemed to be a good indicator of defensive performance which contributes to team performance. These results confirmed the results presented in other studies [13,23] which attested the importance of defensive performance on match results. Our results also show that running activity (relative distance) of backs was significantly greater (p<0.05) during matches. Nevertheless, with the Britannic ranking classification, running activity was greater during negative results, especially for distance covered at high speed/power intensity (p<0.05). These results are similar to those reported in other studies for which, in other team sports, running activity at high intensity was significantly greater for losing teams [35,36,37]. Time spent on defense and the number of defensive line-breaks conceded may require the necessity to realize more high intensity running efforts. This could explain the differences observed between positive and negative results. However, these results demonstrate the complexity of using GPS based data to identify a valid and reliable key performance indicator in RU throughout a season.

The main aim of this study was to identify how 2WL influences the activity/performance in professional RU players throughout a season. Thus, to analyze this influence, and to avoid comparing the inter-individual differences during matches, we chose to apply an individual normalized score (Z-score) based on the mean and the SD of all matches, individualized for each player and for each parameter over the entire season. Individual indicators, normalized by ball-in-play time, permitted to smoothen activity differences induced by the players' position and profile. By using this methodology, we expected to observe the intra-individual fluctuation of performance throughout the season. Therefore, we highlighted the performance peaks and performance drops throughout the season. From this data transformation, a reduction dimension of data was performed in order to summarize running and specific performance. Fig 2 shows that running performance could be synthesized into 2 dimensions. One regrouped total distance, high speed and high-power metabolic distances, while the other dimension included the number of sprints and accelerations. These results partially corroborate those of Weaving et al. [17] who also show that GPS data may be presented by total distance ran and by distance travelled at high-speed. In our study, we used more variables than Weaving et al. [17]. This is probably why the analysis carried out in the present study demonstrated the importance for including the number of very-high intensity efforts (sprint and high acceleration) in WL quantification in addition to "traditional" variables. The data collected for specific skills shows that no variable contains a sufficient linear co-variability that can be resumed by a synthetic index (Fig 3). These results prove that each action analyzed seems to be independent of another and should be studied separately. Indeed, at professional level, specific tasks and player profiles have a significant importance. The individual performance analysis, based on an individual normalized score, demonstrates that physical performance in RU is complex to summarize, especially in terms of sport specific actions. Finally, using a high-dimensional feature for performance identification seems to be relevant for collecting high quantities of relevant information. Difficulties will nevertheless arise during storage, computation and, consequently, on the understanding of the phenomenon.

The final objective of our study was to highlight the influence of 2WL, at short and moderate terms, on individual performance/locomotor activity during matches. The first method (Fig 4) was based on a linear model analyzing the correlation between variables of WL (X) and

activity parameters (Y). The use of this method did not reveal any significant effect of WL on the activity/performance during matches. This first result outlines the limitation of linear models to analyze the interactions between WL and performance. Data mining processes made it possible to reveal significant effects of WL variables on some locomotor activity/performance parameters (Figs 5 & 6 and Table 7). Indeed, data mining processes demonstrated that the number of sprints and high accelerations were negatively influenced when the weighted average of the time spent in low HR intensity (>85% $HR_{max}$) was superior to 218.9 min (Fig 6A). This result emphasizes that too much time spent at low-intensity efforts during training sessions may negatively impact sprinting/accelerating ability. These results are in agreement with those observed in other studies showing the negative effect that training spent in low intensity zones has on the reduction of neuromuscular performance during a professional team-sport season [29,38]. Indeed, Dubois et al. [29] observed significant correlations between % of moderate and high-speed running distances and drop jump testing performance at short term. This demonstrated the negative influence of low-intensity training sessions on neuromuscular performance. Nevertheless, these results do not suggest that training spent at low intensity should be completely ignored. Indeed, during a typical competitive week, the first session of the week (36h after a match) was devoted to technical and tactical training and was performed at low intensity according to a tactical periodization approach [39]. Therefore, the present results seem to show more interest in devoting training to high intensity efforts during other training sessions of the week, even if it means reducing training volume.

In the present study, another significant correlation was observed between chronic load (4 week rolling average), the number of severe impacts (>8G) and the number of successful offensive duels (Fig 6B). In fact, a chronic number of severe impacts greater than 22.6 per week positively impacted this performance parameter. Indeed, the capacity to beat a defender represents an important aspect of offensive performance and contributes to positive team performance [14]. In our study, a greater number of impacts was reached during small-sided training situations. This type of situation, which resembles competitive situations because of increased space-time pressure conditions, enhanced a player's ability to beat his direct opponent. The results concerning the number of severe impacts also illustrates training complexity and the particular difficulty to balance training loads between over-reaching and under-training [1,40]. Indeed, Table 4 shows the negative effects that high quantities of severe impacts during training has on performance at short and moderate terms. This result could be explained by neuromuscular collisions-induced fatigue [32,33]. Dubois et al. [6] also showed a possible negative effect of low exposure to impacts during training sessions on injury rate at short term. However, this study did not specify the effect of this parameter on the types and severity of injuries. Nevertheless, all these results demonstrate the necessity to include specific-training situations including high-intensity actions combined with an "optimal" number of contacts to promote the optimization of individual and team performance. Finally, data mining processes seem to be a "new" method that may contribute to a better understanding of the underlying interactions between practice dosage of locomotor activity/performance in a competitive context [41]. However, despite a high-dimensional approach including an important number of variables, only a few interactions were significantly observed. This indicates that team and individual performance remains difficult to model and identify. Furthermore, the contextual factors (social, psychological, motivational, . . .) were not considered in WL quantification. These factors could interfere in the "dose-response" relationships between training "dose" and physiological adaptations or performance (response). Finally, it would be interesting to study these interactions individually. Indeed, an individual's physical capacity profile may alter how the player copes with the physiological stress induced from practice [8].

## Conclusion and practical applications

The study highlighted the importance of defensive skills for team performance during elite RU matches. Indeed, the number of tackles completed and the number of offensive tackles, especially involving forwards, seemed to be a positive indicator of performance in elite RU, thus corroborating the results of other studies [13,23]. Moreover, forwards presented a greater ($p < 0.05$) activity index (number of coded actions normalized to ball-in-play time) when matches were won, demonstrating the importance of developing a player's ability to repeat high-intensity rugby-specific actions. As for backs, the locomotor activity (GPS data) seems to be an indicator of performance. Nevertheless, all these results must be considered cautiously as they were obtained from analysis based on a single team. Therefore, all these results were largely influenced by the team's tactical and strategical preferences as well as its' mindset. Secondly, this study pointed out that locomotor activity during matches can be summarized by 2 dimensions: one including the total distance travelled, high-speed and high-metabolic running efforts and a second one which corresponds to the number of sprints and fast accelerations. Unfortunately, it was not possible to resume the different specific actions into a synthetic index relating the influence of positional demands and activity profiles in elite rugby players. Finally, the last purpose of this study was to model the influence of WL at different terms (acute, chronic and up to 8 previous weeks) on match performance. The first method based on colinear analysis did not provide significant relationships between WL parameters and performance variables. The use of a threshold-based model, from data mining processes, permitted to identify the influence of WL parameters on different performances variables. Thus, the chronic number of severe impacts seemed to be one of the most influential factors of specific performance, and more specifically on the number of offensive duels won. Therefore, the specific drills/skills including contacts/collisions seems to increase the player's ability to beat the opposition. However, other studies revealed that a high exposure to collision may induce neuromuscular fatigue [6,32,33]. This parameter illustrates perfectly the complexity of training: i.e. how to tune WL as to be between the too much and the not-enough. To conclude, we think that data mining processes will help scientists and sports practitioners develop a better understanding of the underlying relationships between 2WL and match performance. This will undeniably contribute to the ever-striving quest of reaching peak performance by optimizing training processes.

## Acknowledgments

Thank you for the players who have accepted to participated to this study. Finally, a very big thank you to the *SASP Club Athletique Brive Correze* and its medical and technical staff which allowed this study.

## Author Contributions

**Conceptualization:** Romain Dubois, Thierry Paillard, Olivier Maurelli, Jacques Prioux.

**Data curation:** Romain Dubois, Noëlle Bru.

**Formal analysis:** Noëlle Bru.

**Investigation:** Romain Dubois, Jacques Prioux.

**Methodology:** Romain Dubois, Noëlle Bru, Thierry Paillard, Anne Le Cunuder, Jacques Prioux.

**Project administration:** Romain Dubois, Jacques Prioux.

**Supervision:** Thierry Paillard, Jacques Prioux.

**Validation:** Thierry Paillard, Jacques Prioux.

**Visualization:** Thierry Paillard, Olivier Maurelli, Kilian Philippe, Jacques Prioux.

**Writing – original draft:** Romain Dubois, Noëlle Bru.

**Writing – review & editing:** Thierry Paillard, Anne Le Cunuder, Mark Lyons, Kilian Philippe.

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
