## [Decision Letter · Decision Letter 0]

11 Nov 2019

PONE-D-19-23206

Game performances and weekly workload: using of data mining process to enter in the complexity

PLOS ONE

Dear Mr Romain,

Thank you for submitting your manuscript to PLOS ONE. After careful consideration, we feel that it has merit but does not fully meet PLOS ONE’s publication criteria as it currently stands. Therefore, we invite you to submit a revised version of the manuscript that addresses the points raised during the review process.

We would appreciate receiving your revised manuscript by Dec 26 2019 11:59PM. To enhance the reproducibility of your results, we recommend that if applicable you deposit your laboratory protocols in protocols.io, where a protocol can be assigned its own identifier (DOI) such that it can be cited independently in the future. For instructions see: http://journals.plos.org/plosone/s/submission-guidelines#loc-laboratory-protocols

We look forward to receiving your revised manuscript.

Kind regards,

Cristina Cortis, Ph.D.

Academic Editor

PLOS ONE

Journal Requirements:

2. In your original submission form you indicated that ‘Oral permission of ethical comitte of Rennes university’ was obtained form this study, could you provide documentation from Rennes university (e.g. in the form of a letter from the ethics committee) with your revised manuscript confirming the study is exempt from the need for ethical review in your setting?

3. In your data statement you indicate the following ‘However, data access requests from researchers will be possible with the agreement of the club (SASP Club Athletique Brive Correze) by contacting the corresponding author (romain.dubois55@orange.fr).’ please note that we cannot accept having the author as the sole point of contact to access the data, please provide an additional independent contact for data access (e.g. another contact from your institution, but independent from those listed as authors), as well as confirmation that the club agrees to make the data available to other interested researchers if the manuscript is published’.

4. Thank you for stating the following financial disclosure: "The funders had no role in study design, data collection and analysis, decision to publish, or preparation of the manuscript."

Please provide an amended Funding Statement that declares *all* the funding or sources of support received during this specific study (whether external or internal to your organization) as detailed online in our guide for authors at http://journals.plos.org/plosone/s/submit-now

Reviewers' comments:

Reviewer's Responses to Questions

**Comments to the Author**

1. Is the manuscript technically sound, and do the data support the conclusions?

Reviewer #1: Yes

Reviewer #2: Yes

2. Has the statistical analysis been performed appropriately and rigorously? 

Reviewer #1: Yes

Reviewer #2: Yes

3. Have the authors made all data underlying the findings in their manuscript fully available?

Reviewer #1: Yes

Reviewer #2: Yes

4. Is the manuscript presented in an intelligible fashion and written in standard English?

Reviewer #1: Yes

Reviewer #2: Yes

5. Review Comments to the Author

Reviewer #1: This study aimed to identify key performance indicators of professional rugby matches, also analyzing how weekly workload (2WL) can influence match performance throughout an entire season at different time-points.

Introduction

In this section, it has been reported that individual and/or collective performance during RU matches have never been categorized by a key performance indicator (lines 36-38). Actually, literature reported recent and interesting data about KPIs in RU. In particular, the following two studies (Ungureanu et al., 2019a,b) should be considered to better provide the state of the art about this specific topic.

Ungureanu A.N., Brustio P.R., Mattina L., Lupo C. (2019). “How” is more important than “how much” for game possession in elite northern hemisphere rugby union. Biology of Sport. 36(3), 265-272.

Ungureanu A.N., Condello G., Pistore S., Conte D., Lupo C. (2019). Technical and tactical aspects in Italian youth rugby union in relation to different academies, regional tournaments, and outcomes. Journal of Strength and Conditioning Research. 33(6), 1557-1569.

Lines 47-53. Less absolute considerations should be reported about the accuracy (and reliability) of GPS data. In fact, according to Buchheit et al. 2014 (“Monitoring accelerations with GPS in football – Time to Slow Down?”) and Malone et al., 2017 (“Unpacking the Black Box: Applications and Considerations for Using GPS Devices in Sport”), not all parameters downloaded by means of GPS devices are able to provide reliable results and strong funding (especially for time motion parameters such as acceleration, deceleration and changes of direction).

Lines 77-78. “sport-specific activity (tackle count, duels won, …).” the list of technical and tactical parameters should be more exhaustively reported or linked to a table.

Discussion

Lines 359-364. As reported, several factors can determine different outcomes of matches in team sports according to previous comment paper which could be cited (Lupo C., Tessitore A. (2016). How important is the final outcome to interpret match analysis data: the influence of scoring a goal, and difference between close and balance games in elite soccer. Comment on Lago-Penas and Gomez-Lopez. Perceptual & Motor Skills. 122(1). 280-285).

At the end of the Discussion section, same experimental limitations should be reported. In particular, the above mentioned limitations related to the GPS reliability (Lines 47-53) as well as the potential influences of session-RPE applied to game/tactical training in team sports could be reported (Lupo C., Tessitore A., Gasperi L, Gómez MA. (2017). Session-RPE for quantifying the load of different youth basketball training sessions. Biology of Sport. 34, 11-17)

Reviewer #2: This is an interesting new approach to a old question. My criticism is minor.

In Table 2, the intensity and sRPE should use ref 16

In Figure 2, the font should be larger. If the journal photo reduces the size, the text will not be legible. Similarly, in Fig 6, the font is too small.

In Table 3, your first data column lists sRPE. The usual manner of recording would be in RPE units (0-10), and there should be an additional column LOAD, which is the product of sRPE and volume (which in most of the literature is reported in minutes). Likewise, in Table 4, the acute and chronic sRPE that you reoport is more like load (e.g. sRPE x duration). I do find it interesting that you report comparatively modest acute;chronic sRRPE (e.g. LOAD) ratios. This is a topic that has been of some interest, and some controversy, but is potentially an important measure.

6. PLOS authors have the option to publish the peer review history of their article (what does this mean?). If published, this will include your full peer review and any attached files.

Reviewer #1: No

Reviewer #2: No

---

## [Author Response · Author response to Decision Letter 0]

21 Dec 2019

I thought I had met your criteria. I changed the title layout for the figures. After checking I think everything it is ok. However, if something else don’t match, please tell me what's not exactly right. Thank you very much for your help and patience.

2. In your original submission form you indicated that ‘Oral permission of ethical comitte of Rennes university’ was obtained form this study, could you provide documentation from Rennes university (e.g. in the form of a letter from the ethics committee) with your revised manuscript confirming the study is exempt from the need for ethical review in your setting?

I sincerely apologize. I was confused when I noted that this study has been validated by an ethics committee. Indeed, this study was conducted under the control and the approbation of the M2S laboratory (RENNES university laboratory) (see the attached attestation). Moreover, this protocol did not require the opinion of an Ethics Committee because the data collected are those usually collected routinely during training sessions and matches in close cooperation with technical and tactical staffs of the club. As, no biological sampling or psychological assessment was conducted, no prior evaluation by an ethics committee was mandatory (see the attestation of the director of the laboratory involved in this study, attached document).

3. In your data statement you indicate the following ‘However, data access requests from researchers will be possible with the agreement of the club (SASP Club Athletique Brive Correze) by contacting the corresponding author (romain.dubois55@orange.fr).’ please note that we cannot accept having the author as the sole point of contact to access the data, please provide an additional independent contact for data access (e.g. another contact from your institution, but independent from those listed as authors), as well as confirmation that the club agrees to make the data available to other interested researchers if the manuscript is published’.

We have added 2 other contacts, independent from those listed as authors in article. Likewise, we also confirm that the club agrees to make the data available to other interested researchers if the manuscript is published.

4. Thank you for stating the following financial disclosure: "The funders had no role in study design, data collection and analysis, decision to publish, or preparation of the manuscript."

We made the change in cover letter in funding section.

5. Please provide an amended Funding Statement that declares *all* the funding or sources of support received during this specific study (whether external or internal to your organization) as detailed online in our guide for authors at http://journals.plos.org/plosone/s/submit-now

We made the change in cover letter in funding section.

We made the change in cover letter.

7. We note that you have indicated that data from this study are available upon request. PLOS only allows data to be available upon request if there are legal or ethical restrictions on sharing data publicly. For information on unacceptable data access restrictions, please see http://journals.plos.org/plosone/s/data-availability#loc-unacceptable-data-access-restrictions

At the moment, I am a physical coach inside SASP Club Athletique Brive Correze. Contractually, I don’t have the right to share the individual data before June 2021. The club accept, as part of my thesis, I publish collective data but my contract blocks me to make public the individual data before June 2021. If I gave a public access before this date, the club can fine me. Nevertheless, for scientific study I can send the data. 

In the previous paragraphs, I explain why the data cannot be make public before the June 2021. In the same time, I give different contact to have and anticipated access to the data.

Reviewer #1: 

Introduction;

In this section, it has been reported that individual and/or collective performance during RU matches have never been categorized by a key performance indicator (lines 36-38). Actually, literature reported recent and interesting data about KPIs in RU. In particular, the following two studies (Ungureanu et al., 2019a,b) should be considered to better provide the state of the art about this specific topic.

Ungureanu A.N., Brustio P.R., Mattina L., Lupo C. (2019). “How” is more important than “how much” for game possession in elite northern hemisphere rugby union. Biology of Sport. 36(3), 265-272.

Ungureanu A.N., Condello G., Pistore S., Conte D., Lupo C. (2019). Technical and tactical aspects in Italian youth rugby union in relation to different academies, regional tournaments, and outcomes. Journal of Strength and Conditioning Research. 33(6), 1557-1569.

Thank you for your good remark ! We added these references. We also added the Cunningham’s reference which also analysis the key performance indicators in rugby union.

Lines 47-53. Less absolute considerations should be reported about the accuracy (and reliability) of GPS data. In fact, according to Buchheit et al. 2014 (“Monitoring accelerations with GPS in football – Time to Slow Down?”) and Malone et al., 2017 (“Unpacking the Black Box: Applications and Considerations for Using GPS Devices in Sport”), not all parameters downloaded by means of GPS devices are able to provide reliable results and strong funding (especially for time motion parameters such as acceleration, deceleration and changes of direction).

We are fully agreed with this remark. However, we have chosen to present this material limitation in discussion section which is for us, a better place to treat about the material accuracy. We also respond to your last comment in the same time. See Lines 341-344.

Lines 77-78. “sport-specific activity (tackle count, duels won, …).” the list of technical and tactical parameters should be more exhaustively reported or linked to a table.

Thank for this advice. We have chosen to link the table to avoid adding too much text.

Discussion

Lines 359-364. As reported, several factors can determine different outcomes of matches in team sports according to previous comment paper which could be cited (Lupo C., Tessitore A. (2016). How important is the final outcome to interpret match analysis data: the influence of scoring a goal, and difference between close and balance games in elite soccer. Comment on Lago-Penas and Gomez-Lopez. Perceptual & Motor Skills. 122(1). 280-285).

In line 378-379, we underline the importance of contextual factors (score, weather, level of the opposition, …) on game performance and we added the recommended reference.

At the end of the Discussion section, same experimental limitations should be reported. In particular, the above mentioned limitations related to the GPS reliability (Lines 47-53) as well as the potential influences of session-RPE applied to game/tactical training in team sports could be reported (Lupo C., Tessitore A., Gasperi L, Gómez MA. (2017). Session-RPE for quantifying the load of different youth basketball training sessions. Biology of Sport. 34, 11-17)

Thank you for your remark. In our opinion we have already discuss about the factors may affecting the S-RPE session during tactical/technical training session. And the reference which we used, speak about these factors in rugby. Therefore, we estimated that is enough to support our discussion.

Reviewer #2: 

This is an interesting new approach to a old question. My criticism is minor.

In Table 2, the intensity and sRPE should use ref 16

Thank you. That is true. We changed the reference with the good one.

In Figure 2, the font should be larger. If the journal photo reduces the size, the text will not be legible. Similarly, in Fig 6, the font is too small.

We increased the size of the font. Thank you for this good remark.

In Table 3, your first data column lists sRPE. The usual manner of recording would be in RPE units (0-10), and there should be an additional column LOAD, which is the product of sRPE and volume (which in most of the literature is reported in minutes). Likewise, in Table 4, the acute and chronic sRPE that you reoport is more like load (e.g. sRPE x duration). I do find it interesting that you report comparatively modest acute;chronic sRRPE (e.g. LOAD) ratios. This is a topic that has been of some interest, and some controversy, but is potentially an important measure

I think there is a misunderstanding because the S-RPE is “the LOAD”. The S-RPE is obtain by multiplying the RPE quotation in CR-10 Scale by the duration of the session expressed in minutes, in accordance with Foster et al. (2001) and as explain in the table. To avoid a misunderstanding, we suppressed the intensity variable because we did not use in our results and discussion, hoping that allow to increase the clarity about this point.

Likewise, we made the choice to keep in the table, the volume duration expressed in hour to give at the lecturer a pract

---

## [Decision Letter · Decision Letter 1]

8 Jan 2020

Game performances and weekly workload: using of data mining process to enter in the complexity

PONE-D-19-23206R1

Dear Dr. Romain,

We are pleased to inform you that your manuscript has been judged scientifically suitable for publication and will be formally accepted for publication once it complies with all outstanding technical requirements.

With kind regards,

Cristina Cortis, Ph.D.

Academic Editor

PLOS ONE

Additional Editor Comments (optional):

Reviewers' comments:

Reviewer's Responses to Questions

**Comments to the Author**

1. If the authors have adequately addressed your comments raised in a previous round of review and you feel that this manuscript is now acceptable for publication, you may indicate that here to bypass the “Comments to the Author” section, enter your conflict of interest statement in the “Confidential to Editor” section, and submit your "Accept" recommendation.

Reviewer #1: All comments have been addressed

Reviewer #2: All comments have been addressed

2. Is the manuscript technically sound, and do the data support the conclusions?

Reviewer #1: Yes

Reviewer #2: Yes

3. Has the statistical analysis been performed appropriately and rigorously? 

Reviewer #1: Yes

Reviewer #2: Yes

4. Have the authors made all data underlying the findings in their manuscript fully available?

Reviewer #1: Yes

Reviewer #2: Yes

5. Is the manuscript presented in an intelligible fashion and written in standard English?

Reviewer #1: Yes

Reviewer #2: Yes

6. Review Comments to the Author

Reviewer #1: (No Response)

Reviewer #2: all of my comments have been adequately addressed. I have no other comments relative to this paper.

7. PLOS authors have the option to publish the peer review history of their article (what does this mean?). If published, this will include your full peer review and any attached files.

Reviewer #1: No

Reviewer #2: No

---

## [Editor Report · Acceptance letter]

15 Jan 2020

PONE-D-19-23206R1 

Game performances and weekly workload: using of data mining process to enter in the complexity 

Dear Dr. Romain:

I am pleased to inform you that your manuscript has been deemed suitable for publication in PLOS ONE. Congratulations! Your manuscript is now with our production department. 

With kind regards,

on behalf of

Dr. Cristina Cortis 

Academic Editor

PLOS ONE